# Diagnosis of Metastatic Non-Small Cell Lung Cancer during Hospitalization: Missed Opportunity for Optimal Supportive Care?

**DOI:** 10.3390/cancers16061221

**Published:** 2024-03-20

**Authors:** Shristi Upadhyay Banskota, Jonathan Q. Trinh, Elizabeth Lyden, Conor Houlihan, Samia Asif, Omar Abughanimeh, Benjamin A. Teply

**Affiliations:** Division of Hematology and Oncology, Department of Internal Medicine, University of Nebraska Medical Center, Omaha, NE 68198, USA; supadhyaybanskota@unmc.edu (S.U.B.); elyden@unmc.edu (E.L.); choulihan@unmc.edu (C.H.); samia.asif@unmc.edu (S.A.); omar.abughanimeh@unmc.edu (O.A.); ben.teply@unmc.edu (B.A.T.)

**Keywords:** metastatic non-small cell lung cancer, inpatient, palliative care

## Abstract

**Simple Summary:**

Many of the current data on outcomes in metastatic non-small cell lung cancer (NSCLC) stem from patients who were diagnosed in the ambulatory setting. This study aimed to investigate patients who receive new diagnoses of metastatic NSCLC while they are inpatients. Specifically, we sought to evaluate how many patients receive systemic therapy as well as the overall population’s survival outcomes and palliative care utilization in order to help guide treatment for future cases.

**Abstract:**

Purpose: The usual workup for patients newly diagnosed with advanced non-small cell lung cancer (NSCLC) occurs in the ambulatory setting. A subset of patients present with acute care needs and receive the diagnosis while hospitalized. Palliative therapies are typically initiated when patients are outpatients, even when diagnoses are made when they are inpatients. Lengthy admission, rehabilitation needs after discharge, and readmissions are possible barriers to timely and adequate outpatient follow-up. The outcomes for these patients diagnosed in the hospital are not well characterized. We hypothesized that patients have been ill-served by current treatment patterns, as reflected by low rates of cancer-directed treatment and poor survival. Patients and methods: We performed a retrospective study of new inpatient diagnoses of metastatic NSCLC at our institution between 1 January 2012 and 1 January 2022. The primary outcome was the proportion of patients ultimately receiving cancer-directed therapy. Other outcomes included time to treatment, use of targeted therapy, palliative care/hospice utilization, and overall survival (OS). Results: Seventy-three patients were included, with a median age of 57 years. Twenty-seven patients (37%) ultimately received systemic therapy with a median time from diagnosis to treatment of 37.5 days. Overall, 5.4% patients died while admitted, 6.8% were discharged to a hospice, 21.9% were discharged to a facility, and 61.6% were discharged home. Only 20 patients (27%) received palliative care consultation. The median OS for our entire population was 2.3 months, with estimated 6-month and 1-year OS rates of 32% and 22%, respectively. Conclusion: Patients with new inpatient diagnoses of metastatic NSCLC have extremely poor outcomes. Current management strategies resulted in few patients starting systemic therapy, yet most of the patients did not receive palliative care or hospice involvement. These findings demonstrate that there is a high unmet need to optimally support and palliate these patients.

## 1. Introduction

The workup for the diagnosis of solid tumors is typically performed in the ambulatory setting [1,2,3]. Often, this process begins with screening programs or symptomatic patients who present to general practitioners, leading to referrals to specialists. As a result, most of the data regarding solid tumor outcomes are rooted in this population, and the applicability of these data to those with new diagnoses in acute care settings is unknown. For the latter, the standard of care for most patients is to initiate systemic therapy after the acute care needs have resolved.

This current approach is based on poor outcomes in patients who start systemic therapy in acute care settings. For example, although the role of immune checkpoint inhibition (ICI) has grown substantially in recent years, nearly a quarter of patients with advanced disease who receive inpatient ICI treatment die during the same admission, and about half die within 30 days of treatment [4]. Similarly, high rates of early mortality occur in patients who receive inpatient chemotherapy [5,6]. An inferior performance status in the acutely ill likely plays a significant role in these poor outcomes. Patients with lower Karnofsky performance status (KPS) scores have a worse predicted prognosis, and, by definition, impairment indicating hospital admission results in a KPS score of 30% on a scale from 0% to 100% [7,8,9,10,11]. In addition, inpatient administration of systemic therapy incurs a high financial burden on both patients and the health care system [7].

Lung cancer remains the leading cause of cancer deaths worldwide [12]. Non-small cell lung cancer (NSCLC) comprises approximately 84% of all lung cancers [13]. Though decreasing, its annual incidence rate in the United States is still above 40 per 100,000 persons. Moreover, about half of new diagnoses are stage IV, with 1- and 5-year survival rates of approximately 31.3% and 5.8%, respectively. A paucity of data exists regarding patients with advanced NSCLC who are diagnosed while they are inpatients, a population that poses a particularly challenging clinical dilemma as the evolution in treatment options for NSCLC has led to possible benefits even in those with a poor performance status [14,15,16,17].

Given this uncertainty regarding the optimal approach for management, these patients may be at risk of poorly coordinated or delayed end-of-life care. In order to address this predicament, we must first explore outcomes in patients who are diagnosed with metastatic NSCLC while they are inpatients, who are not well characterized at this time. We hypothesize that, regardless of whether or not systemic therapy is pursued, these patients have poor clinical outcomes measured not only by their overall survival, but also by a lack of palliative care involvement. To investigate this, we analyzed all patients with new inpatient diagnoses of metastatic NSCLC over a ten-year period at our institution.

## 2. Methods

This was a retrospective study of patients with newly diagnosed advanced-stage NSCLC during hospitalization. We studied patients who were admitted to the University of Nebraska Medical Center (UNMC) between January 2012 and January 2022 and were diagnosed with advanced-stage NSCLC during the same admission. Our primary outcome was determining the proportion of patients ultimately receiving cancer-directed therapy. Some additional outcomes were time to treatment initiation, the utilization of targeted therapy, palliative care/hospice utilization, overall survival (OS), and time to outpatient follow-up. The study was approved by the Institutional Review Board of UNMC.

To obtain the study population, we screened patients who underwent inpatient biopsies between January 2012 and January 2022 using current procedural terminology (CPT) codes. We selected the procedure codes for the following biopsy sites: lung, mediastinum, pleura, liver, lymph node, bone, brain, and mediastinum. We then cross-referenced the group of patients with ICD 10 codes for lung cancer, namely C34, which stands for “Malignant neoplasm of bronchus and lung”.

Once we received access to the group of study patients, we extracted the data from the electronic medical record into Microsoft Excel. We performed data analysis using SAS version 9.4. We used counts and percentages to describe categorical data and used medians and ranges to summarize continuous data. Additionally, we estimated overall survival (OS) using the Kaplan–Meier method. We defined overall survival (OS) as the time from diagnosis to last contact or death from any cause. Additionally, we used a multivariable Cox regression model to determine the effect of treatment status on mortality risk while adjusting for the other covariates of interest, such as age, gender, histology, smoking status, status of brain metastases, and disposition. For this multivariable analysis, we excluded patients who died during admission or were discharged to a hospice in order to properly compare treatment status.

## 3. Results

Our initial screening included a total of 200 patients who received a new diagnosis of metastatic NSCLC from 1 January 2012–1 January 2022. However, only 73 NSCLC cases were eventually included in the analysis (Table 1). We excluded 127 cases for one or more of the following reasons: the patient had a prior earlier stage of NSCLC that progressed to stage IV NSCLC; the diagnosis of lung cancer was established when the patient was an outpatient and prior to the hospitalization; or the diagnostic workup was initiated while the patient was an outpatient and a biopsy was performed during admission for alternative causes. The median age was 64 years, and the median BMI was 25.2 kg/m^2^. Overall, 40 patients (54.8%) were male, 61 (83.6%) were Caucasian, and 9 (12.3%) had never smoked.

The most common chief complaint on admission was shortness of breath (37.0%), while the most common histology was adenocarcinoma (49.3%). A relatively low proportion of patients received a formal oncology consultation during their admission (23 patients, 31.5%), while a higher percentage of patients had their first meeting with an oncologist in the outpatient setting (35 patients, 47.9%). The median time from diagnosis to the former was 1 day, as opposed to 17 days for the latter.

We identified 24 (32.8%) patients with brain metastases, and 60 (82.1%) patients underwent next-generation sequencing (NGS) testing. Surprisingly, only 20 (27.4%) patients received a palliative care consultation while they were an inpatient. Regarding discharge, 61.6% of patients were discharged home, 21.9% of patients were discharged to a facility, and 5.4% died while they were an inpatient. Twenty-four (34.8%) patients were readmitted in the 30 days following discharge, most commonly due to shortness of breath.

Ultimately, 27 (37%) patients underwent systemic treatment with a median time of 37.5 days from diagnosis to initiation of treatment. Of these 27, 22 received chemotherapy with or without immunotherapy (Figure 1). Three patients received immunotherapy alone, and two patients received epidermal growth factor receptor (EGFR) tyrosine kinase inhibitors (TKI’s) based on molecular studies.

The median overall survival (OS) of our entire patient population was extremely poor, at 2.3 months (Figure 2). The estimated 6-month and 12-month OS rates were 32% and 22%, respectively. Survival did not significantly differ between patients who were discharged to home versus a facility (2.7 vs. 2 months). Fifty-six (76.7%) patients ultimately died. However, only 23 (31.5%) were enrolled in a hospice at any time (Table 1).

In our multivariate analysis, we found that treatment status was associated with survival after adjusting for other variables of interest in the model (*p* < 0.0001). The risk of death for patients who did not receive treatment was found to be 13.85 times the risk of death for patients who ultimately received treatment for their disease (Table 2).

## 4. Discussion

In our study, patients with new inpatient diagnoses of metastatic NSCLC had extremely poor outcomes, with a median survival of 2.3 months. This is in contrast to the current revolution seen in survival in clinical trials among patients with advanced-stage NSCLC [18,19,20]. The identification of actionable molecular alterations and the increasing utilization of novel therapies such as targeted therapies and immunotherapies have transformed the treatment landscape of advanced-stage NSCLC [21,22]. However, in the current era of precision oncology, it is crucial to understand which subgroup of patients will benefit the most from our current standard of care treatments rather than using the “one size fits all” approach.

Upon evaluation of the existing literature, we found a paucity of data providing insights into the impact of diagnosis setting on patient outcomes to eventually guide our clinical decision making.

In one prior real-world single-institution study, Gotfrit et al. examined newly diagnosed stage IIIb or IV NSCLC patients whose initial oncology consultation was performed while they were an inpatient [23]. The median OS for this population was 2.1 months, comparable to the 2.3 months in our population. The authors performed another study later comparing those who met with medical oncology as inpatients with those who did so as outpatients [24]. Only 21% of the inpatient cohort ultimately received systemic therapy, as opposed to 55% of the outpatient cohort. The inpatient treatment group had a shorter median overall survival compared to the outpatient treatment group (8.4 vs. 10.5 months). However, those that received treatment had interestingly higher response rates than their outpatient counterparts [24]. Our study included only new inpatient diagnoses of stage IV NSCLC, which likely contributed to the lower overall survival in our patients. The patients in our study who ultimately received cancer-directed systemic therapies only did so in an outpatient setting after discharge from the hospital. Barth et al. evaluated the characteristics and outcomes of new inpatient diagnoses of metastatic lung cancer, but only included patients admitted to intensive care units. Additionally, they included both small cell and non-small cell lung cancer patients [25].

Under the current guidelines, the management of advanced stage non-small cell lung cancer usually comprises next-generation sequencing studies (NGS) and analysis of the PD-L1 status of the tumor at diagnosis [26]. Treatment decisions are then made in the outpatient setting once the results of molecular studies are available, which can take a few weeks on average [27]. In our study group where patients received treatments based on standard guidelines, only 27% of them ultimately received some form of systemic therapy. Only 3% (*n* = 2) of patients received targeted therapy; this may be due to the fact that 51% of our patients did not have adenocarcinoma. This may also reflect our timeline, since guidelines did not recommend EGFR testing for all patients with adenocarcinoma until 2012 [28]. Additionally, recent advances in diagnostic tools, such as fine needle aspiration cytology, have allowed for the more rapid and less invasive evaluation of molecular testing of tumors compared to years prior [29].

The benefit of early palliative care enrollment for patients with metastatic NSCLC is well established, including for those who were diagnosed as inpatients [30,31]. This includes a better quality of life, mood, and overall survival. Despite this, only 27.4% of our patients received a palliative care consultation while they were inpatients, and most patients never enrolled in a hospice at any point. Given the short survival period, low rate of systemic therapy, and rare involvement of palliative care, a disconnect in optimizing patient care seems to exist that must be addressed.

Through our multivariate analysis, we demonstrated that while the whole population had poor outcomes, the small group of patients that ultimately achieved an outpatient status and received cancer-directed treatment had better survival outcomes compared to those who never received treatment. This finding is comparable with general outpatient diagnoses of metastatic non-small cell cancer. However, this raises the major question: will most inpatient diagnoses with advanced non-small cell cancer ever make it to the outpatient setting and be fit enough to receive cancer-directed treatment? Based on our study findings, they often do not. Hence, we believe the answer regarding what to do with these inpatient diagnoses is more nuanced and some patients indeed end up missing out on the optimal end-of-life care in this dilemma.

It is also important to realize certain limitations of the present study. As a retrospective study, the results are observational in nature. We also cannot comment on certain nuances of care, such as if patients were offered palliative care but declined or engaged in daily discussions about end-of-life care during hospitalizations. Additionally, we examined a relatively small sample size at a single institution in Nebraska. Though our institution serves patients from multiple states, it still represents a small portion of the general population. In 2023, Nebraska had an estimated 1340 new cases of lung cancer and an estimated 630 deaths from lung cancer [32]. For more generalizable results, a multicenter study would be beneficial.

In conclusion, this study demonstrates the very poor prognosis of patients who receive new diagnoses of stage IV NSCLC while they are inpatients. More importantly, it reveals the potential need to focus more on palliative care and less on aggressive treatment in this patient population.

Some factors contributing to the adverse prognosis in this group of patients could be the more extensive disease burden at diagnosis secondary to the delayed detection and diagnosis of the cancer itself. This might often contribute to an already advanced stage at presentation and might suggest an aggressive disease biology. Other additional factors such as a higher comorbidity burden and compromised overall health with poor performance status might also negatively impact treatment tolerability and overall survival in these patients. Future larger studies to enhance our understanding of outcome differences in metastatic NSCLC are crucial to guide medical oncologists in patient-focused care and medical decision making.

## 5. Conclusions

Patients diagnosed with metastatic NSCLC while they are inpatients represent an understudied yet clinically significant patient population. These patients generally have extremely poor survival outcomes, despite the evolution of treatment options available. While those who do ultimately receive systemic therapy may benefit from it, it is also vital to involve other resources such as palliative care early and more often.

## Figures and Tables

**Figure 1 cancers-16-01221-f001:**
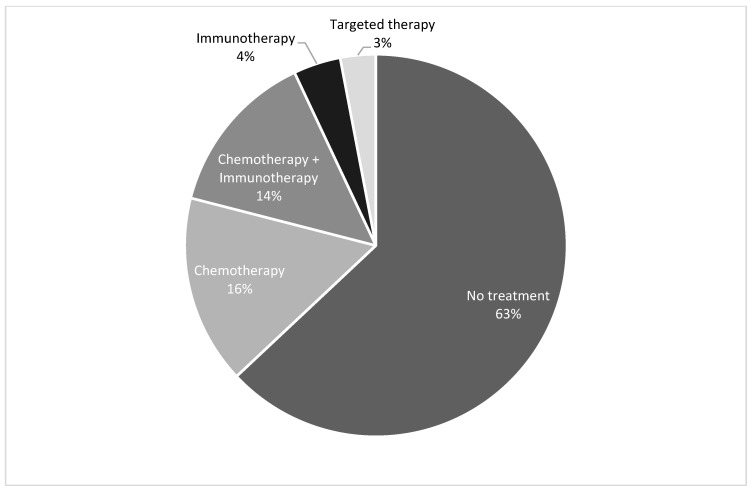
Systemic treatments received by patients in our cohort.

**Figure 2 cancers-16-01221-f002:**
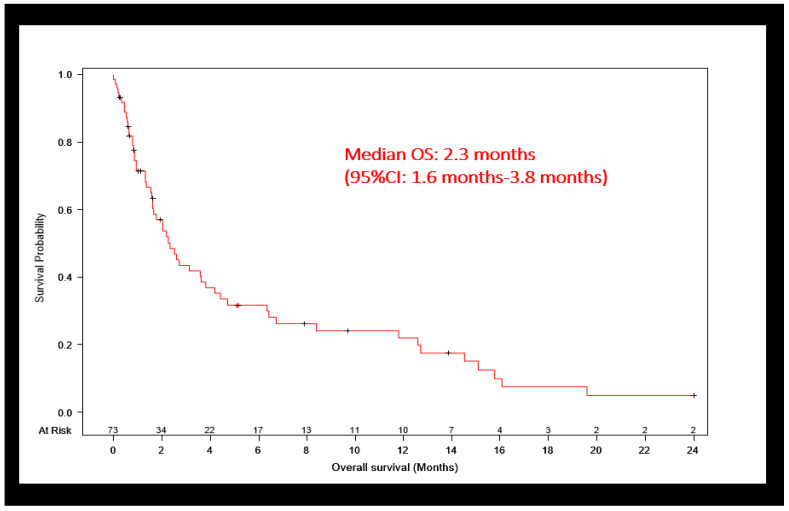
Kaplan–Meier survival curve for our cohort.

**Table 1 cancers-16-01221-t001:** Patient characteristics and hospital courses for our cohort.

Characteristic	Findings
Number of patients	73
Median age (min, max)	64 years (32, 85)
Median BMI (min, max)	25.2 kg/m^2^ (12.2, 46.8)
Gender, *n* (%)	
Male	40 (54.8%)
Female	33 (45.2%)
Race, *n* (%)	
Caucasian	61 (83.6)
African American	7 (9.6%)
Other	5 (6.8%)
Smoking status, *n* (%)	
Current	32 (43.8%)
Former	32 (43.8%)
Never	9 (12.3%)
Histology, *n* (%)	
Adenocarcinoma	33 (49.3%)
Squamous cell	14 (20.9%)
Other	26 (35.6%)
Chief complaint, *n* (%)	
Shortness of breath	27 (37.0%)
Pain	14 (19.2%)
Weakness	12 (16.4%)
Other	20 (27.3%)
Brain metastases, *n* (%)	
Yes	24 (32.8%)
No	39 (53.4%)
Not examined	10 (13.6%)
NGS performed, *n* (%)	
Yes	60 (82.1%)
No	13 (17.8%)
Oncology consult, *n* (%)	
Inpatient	23 (31.5%)
Outpatient	35 (47.9%)
None	15 (20.5%)
Disposition, *n* (%)	
Home	45 (61.6)
Facility	16 (21.9%)
Hospice	5 (6.8%)
Deceased	4 (5.4%)
Against medical advice	3 (4.1%)
Readmitted in next 30 days, *n* (%)	
Yes	24 (32.8%)
Reason for readmission, *n* (%)	
Shortness of breath	11 (45.8%)
Fall	2 (8.3%)
Other	11 (45.8%)
No	49 (67.1%)
Hospice enrollment	
At any time	23 (31.5%)
During initial admission	5
No	50 (68.5%)

**Table 2 cancers-16-01221-t002:** Multivariate analysis of prognostic factors for survival.

Parameter	Parameter Estimate (Standard Error)	Hazard Ratio (95% Confidence Interval)	*p*-Value
Treatment: No treatment vs. Treatment	2.629 (0.507)	13.854 (5.131, 37.405)	<0.0001
Age: 1 year increment	0.004 (0.022)	1.004 (0.961, 1.049)	0.085
Gender: Male vs. Female	0.213 (0.352)	1.238 (0.621, 2.466)	0.545
Histology: Adenocarcinoma vs. Other	−0.322 (0.383)	0.725 (0.342, 1.535)	0.401
Histology: Squamous cell vs. Other	0.046 (0.449)	1.048 (0.434, 2.526)	0.918
Smoking: Quit vs. No	−0.042 (0.589)	0.958 (0.302, 3.038)	0.943
Smoking: Yes vs. no	0.038 (0.595)	1.038 (0.324, 3.331)	0.949
Brain metastasis: Unknown vs. No	−0.611 (0.660)	0.543 (0.149, 1.980)	0.355
Brain metastasis: Yes vs. No	−0.912 (0.375)	0.402 (0.192, 0.839)	0.015

## Data Availability

The raw data supporting the conclusions of this article will be made available by the authors on request.

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
