# Peer review of "Diagnosis of Metastatic Non-Small Cell Lung Cancer during Hospitalization: Missed Opportunity for Optimal Supportive Care?"

_cancers, 2024, doi:10.3390/cancers16061221_

Round 1

Reviewer 1 Report

Comments and Suggestions for Authors

The Authors have addressed utilization of palliative care in metastatic NSCLC patients newly diagnosed when hospitalized. They also aim to evaluate overall survival in this subset. The manuscript is very well written and data clearly presented.  However, there is a lack of novelty, and the focus is somehow unclear. For instance, primary outcome was proportion of patients ultimately receiving cancer-directed therapy; but simple summary states how often resources such as palliative care are utilized.

This is a single institution experience, the Authors could also provide some information about the area and population this institution offers cancer care, including number of newly diagnosed lung cancer cases. It would be helpful to discuss whether OS for non-metastatic and metastatic NSCLC patients diagnosed in outpatient is available from their own centre. Currently, the Authors compare their results with clinical trial data.  More relevant would be comparison with their own data or real world evidence from other centres.

Its relatively long period, 2012-2022, over 10-years period, the number of eligible patients was quite small (73).  Although this has been addressed by the Authors.

Author Response

“The Authors have addressed utilization of palliative care in metastatic NSCLC patients newly diagnosed when hospitalized. They also aim to evaluate overall survival in this subset. The manuscript is very well written and data clearly presented.  However, there is a lack of novelty, and the focus is somehow unclear. For instance, primary outcome was proportion of patients ultimately receiving cancer-directed therapy; but simple summary states how often resources such as palliative care are utilized.

       Thank you for your comments.  In order to clarify the focus of the paper, we have edited the simple summary to precisely align with the paper.  While we appreciate that this is not completely novel, there remains minimal data in the literature regarding these real world outcomes in special subpopulations.

This is a single institution experience, the Authors could also provide some information about the area and population this institution offers cancer care, including number of newly diagnosed lung cancer cases. It would be helpful to discuss whether OS for non-metastatic and metastatic NSCLC patients diagnosed in outpatient is available from their own centre. Currently, the Authors compare their results with clinical trial data.  More relevant would be comparison with their own data or real world evidence from other centres.

       In order to provide additional context for our population at our center, we have added information regarding our patient population in Nebraska, specifically discussing the annual incidence and mortality regarding lung cancer. Additionally, we added a new reference to a similar real-world study that evaluated survival in patients with advanced NSCLC whose initial oncologic consults were done inpatient. To our knowledge, we don’t believe there is other real-world evidence besides that described in our paper.

Its relatively long period, 2012-2022, over 10-years period, the number of eligible patients was quite small (73).  Although this has been addressed by the Authors.”

As the reviewer states, we have addressed the time duration of the study in our discussion.

Reviewer 2 Report

Comments and Suggestions for Authors

I have only minor suggestions.

The study deals with an extended time period during which regulations and guidelines have changed multiple times as new molecular tests and new drugs have become available. Only a minority of patients received targeted therapy (EGFR-TKIs). I think this aspect requires much more discussion for its important implications and clarity to the readers, namely:
- the extended time period likely includes patients that were tested for only a few markers (e.g. early 2010s) or that had histologies for which testing was not indicated (Squamous cell carcinoma).
- biomarkers used to require histological samples for testing. Nowadays we know that cytological samples can be utilised fruitfully (see e.g. https://doi.org/10.1016/j.prp.2021.153547 ) and these cytological samples are especially indicated in patients with low performance status and metastases (stage IV) since a simple fine-needle aspiration biopsy of a metastatic lymph node can give all the prognostic-predictive information required to start targeted therapy in a very short time (even same-day if PCR is employed).

Figure 1 is a bit eccentric and surely not color-blind friendly; I would employ a more standard color scheme.

The abstract could be enhanced by providing specific numerical values for the outcomes mentioned.

Comments on the Quality of English Language

English is fine. Minor typos (e.g. L66: "end-of-life" should be hyphenated since it is used as an adjective; L102, 114, 116, ⋯: sentences should not start with a number)

Author Response

“The study deals with an extended time period during which regulations and guidelines have changed multiple times as new molecular tests and new drugs have become available. Only a minority of patients received targeted therapy (EGFR-TKIs). I think this aspect requires much more discussion for its important implications and clarity to the readers, namely:
- the extended time period likely includes patients that were tested for only a few markers (e.g. early 2010s) or that had histologies for which testing was not indicated (Squamous cell carcinoma).
- biomarkers used to require histological samples for testing. Nowadays we know that cytological samples can be utilised fruitfully (see e.g. https://doi.org/10.1016/j.prp.2021.153547 ) and these cytological samples are especially indicated in patients with low performance status and metastases (stage IV) since a simple fine-needle aspiration biopsy of a metastatic lymph node can give all the prognostic-predictive information required to start targeted therapy in a very short time (even same-day if PCR is employed).

       We very much agree that standards of care for biomarker testing in NSCLC has been rapidly involving, including over the time course of this study.  While in theory, same-day PCR could provide very rapid information for whether targeted therapy can be employed, our NCI-designed tertiary referral Cancer Center (similar to most others) does not have access to CLIA-certified rapid testing.  We have expanded our discussion regarding the biomarker testing in order to highlight this issue.

Figure 1 is a bit eccentric and surely not color-blind friendly; I would employ a more standard color scheme.

We have revised Figure 1 as requested.

The abstract could be enhanced by providing specific numerical values for the outcomes mentioned.

       We have edited the abstract to include specific numerical values for the outcomes.

English is fine. Minor typos (e.g. L66: "end-of-life" should be hyphenated since it is used as an adjective; L102, 114, 116, ⋯: sentences should not start with a number)”